# Muscle architecture, voluntary activation, and low-frequency fatigue do not explain the greater fatigue of older compared with young women during high-velocity contractions

**Liam F. Fitzgerald**[◎]*, **Margaret M. Ryan**[‡], **Miles F. Bartlett**[‡], **Jules D. Miehm**[‡], **Jane A. Kent**[◎]

Department of Kinesiology, University of Massachusetts Amherst, Amherst, Massachusetts, United States of America

◎ These authors contributed equally to this work.
‡ These authors also contributed equally to this work.
* l.fitzgerald@phhp.ufl.edu

**Data Availability Statement:** All relevant data are within the manuscript and its Supporting information files.

## Abstract

Although high-velocity contractions elicit greater muscle fatigue in older than young adults, the cause of this difference is unclear. We examined the potential roles of resting muscle architecture and baseline contractile properties, as well as changes in voluntary activation and low-frequency fatigue in response to high-velocity knee extensor work. Vastus lateralis muscle architecture was determined in quiescent muscle by ultrasonography in 8 young (23.4±1.8 yrs) and 8 older women (69.6±1.1). Maximal voluntary dynamic (MVDC) and isometric (MVIC), and stimulated (80Hz and 10Hz, 500ms) isometric contractions were performed before and immediately after 120 MVDCs ($240°·s^{-1}$, one every 2s). Architecture variables did not differ between groups ($p \geq 0.209$), but the half-time of torque relaxation ($T_{1/2}$) was longer in older than young women at baseline (151.9±6.0 vs. 118.8±4.4 ms, respectively, $p = 0.001$). Older women fatigued more than young (to 33.6±4.7% vs. 55.2±4.2% initial torque, respectively; $p = 0.004$), with no evidence of voluntary activation failure (ΔMVIC:80Hz torque) in either group ($p \geq 0.317$). Low-frequency fatigue (Δ10:80Hz torque) occurred in both groups ($p < 0.001$), as did slowing of $T_{1/2}$ ($p = 0.001$), with no differences between groups. Baseline $T_{1/2}$ was inversely associated with fatigue in older ($r^2 = 0.584$, $p = 0.045$), but not young women ($r^2 = 0.147$, $p = 0.348$). These results indicate that differences in muscle architecture, voluntary activation, and low-frequency fatigue do not explain the greater fatigue of older compared with young women during high-velocity contractions. The inverse association between baseline $T_{1/2}$ and fatigue in older women suggests that factors related to slower muscle contractile properties may be protective against fatigue during fast, repetitive contractions in aging.

## Introduction

Maximal torque and power production are lower in older compared with younger adults [1–6], due to factors such as a loss of muscle mass [5, 7–10], slower maximal contraction velocities

**Funding:** The authors received no specific funding for this work.

**Competing interests:** The authors have declared that no competing interests exist.

[1, 2, 11], and lower maximal motor unit discharge rates [12]. These changes have significant consequences for older adults because deficits in muscular power are associated with an increased likelihood of falling, which may place older adults at risk of injury, poor mobility and subsequent loss of independence [13, 14]. In particular, power production by the knee extensor muscles is important due to its association with physical function in aging [15], as well as the concept that rapid contraction of the knee extensors appears to be crucial for preventing falls following a balance perturbation [16].

The decline in maximal torque and power with old age might be explained, at least in part, by changes in muscle architecture such as muscle thickness, pennation angle, and fascicle length [17]. Generally, muscles with a greater pennation angle have more sarcomeres in parallel, whereas muscles with longer fascicles contain more sarcomeres in series [18]. Conceptually, a greater number of sarcomeres in parallel allows for greater torque production, while more sarcomeres in series allows for greater velocity of shortening [19].

In aging, the loss of muscle mass results in architectural remodeling [20]; fascicles can become shorter and less pennate [21]. Previous work has shown fascicle length and pennation angle of the vastus lateralis to be ~12% shorter and ~25% smaller, respectively, in older compared with young men [22]. Given the apparent lack of sex-related differences in the decline of quadriceps muscle volume with age [8], similar age-related changes in muscle architecture may be expected in women. However, Kubo et al. [23] observed an age-related decrease in fascicle length in the vastus lateralis in men but not women. Changes in fascicle length may account for ~20% of the age-related difference in maximal shortening velocity of the medial gastrocnemius in men [4]. Thus, age-related differences in fascicle length could place older muscles at a disadvantage for generating power during high-velocity contractions [5, 11, 24–26]. To date, the extent to which architectural remodeling affects age-related differences in muscle fatigue is not known.

Muscle fatigue is defined as the reduced capacity to produce torque or power in response to a period of contractile activity [27]. Mechanisms of fatigue can include transient impairments at any site along the pathway of force or power production, from the brain to the myofilaments [27]. The literature generally supports the observation of less muscle fatigue in older ($\geq$65 years) compared with young ($\leq$40 years) adults in response to isometric contractions [28]. Conversely, during dynamic contractions at high velocities, or when fatigue is expressed as a decline in power, older adults fatigue to a greater extent than young adults [5, 11, 24–26, 29]. Despite consistent observations of these age-related differences in muscle fatigue, the mechanisms for greater fatigue in older compared with young adults in response to high-velocity contractions remain poorly understood.

Use of percutaneous stimulation offers a non-invasive approach for identifying whether the cause of fatigue is proximal or distal to the stimulating electrodes (i.e., central or peripheral) [30]. Small reductions (~5%) in voluntary activation, defined as the neural drive to the contracting muscle, have been observed in both young and older men following contractions of the knee extensors at slow- and moderate-velocities [24]. More recently, Sundberg et al. [29] observed failure of voluntary activation in older, but not young, women following 80 maximal-velocity contractions at a load of 20% maximal isometric torque. However, the magnitude of the change in voluntary activation in the older women was small (<2%), and no association was observed between fatigue (% initial power) and changes in voluntary activation. Thus, age-related differences in voluntary activation failure do not appear to contribute to age-related differences in fatigue in response to high-velocity contractions.

The presence of low-frequency fatigue (LFF) can also be evaluated *in vivo* using myoelectric stimulation by comparing fatigue-induced changes in torque in response to low- vs. high-frequency stimuli [31–34]. With LFF, the decline and slowed recovery of torque produced by

low-frequency stimuli are more pronounced than those elicited at higher frequencies [31]. Because LFF may stem, at least in part, from a failure in excitation contraction coupling (ECC) the low:high frequency torque ratio has been used as an indirect measure of ECC failure *in vivo*. In single muscle fibers, a greater loss of force in response to low- vs. high-frequency stimuli is associated with impaired $Ca^{2+}$ release [35, 36]. Thus, age-related impairments in $Ca^{2+}$ handling could contribute to the development of LFF [37]. In response to fatiguing isometric contractions in humans, there appear to be no age-related differences in LFF [32, 33]. However, the potential role of LFF in the greater fatigue and delayed recovery of power in older compared with younger adults in response to high-velocity contractions is not known.

In unfatigued muscle, the half-time of torque relaxation ($T_{1/2}$) is generally longer in older than young adults [5, 24, 25, 29, 38]. This difference could result from greater type 1 fiber composition [39] or differences in cross-bridge kinetics [40] in older muscle; the source of this slowing has not been definitively identified. In fatigue, $T_{1/2}$ lengthens [38, 41], presumably due to a slowing of $Ca^{2+}$ release from troponin or its re-uptake by the sarcoplasmic reticulum [42] in response to the accumulation of $H^+$ [43]. During fatiguing isometric contractions, some authors report a similar slowing of $T_{1/2}$ in young and older muscle [25, 38, 41]. In contrast, during dynamic contractions of the knee extensor muscles, others reported a greater slowing of $T_{1/2}$ in older compared with young men following contractions at moderate and fast, but not slow, velocities [24]. More recently, intracellular pH was shown to be lower in older than young muscle in response to dynamic contractions of the knee extensors to fatigue [44], which could contribute to a greater slowing of $T_{1/2}$ in older than young muscle.

Given the existing questions about the causes of muscle fatigue in aging, the purpose of this study was to compare in young and older women several potential mechanisms that may contribute to age-related differences in fatigue following high-velocity concentric contractions of the knee extensor muscles. Baseline muscle architecture, as well as changes in voluntary activation and contractile properties, were evaluated during fatigue and for 30 minutes of recovery. We hypothesized that fatigue (fall of peak power relative to baseline) would be greater, and recovery of power slower, in older compared with young muscle. We further hypothesized that vastus lateralis fascicle length and pennation angle would be shorter and smaller, respectively, in older compared with young women, and that fascicle length would be positively associated with peak power production at baseline, given the importance of the number of sarcomeres in series for power generation [4]. Based on the current literature, we did not expect to observe any age-related differences in voluntary activation at the end of the fatigue protocol [24, 29]. However, we did anticipate greater low-frequency fatigue and slowing of $T_{1/2}$ in older compared with young women.

## Materials and methods

### Participants

The University of Massachusetts Amherst Institutional Review Board approved this study (approval number: 2015–2413). Prior to enrollment, each participant gave their written, informed consent, as approved by the University of Massachusetts Amherst, and in accordance with the Declaration of Helsinki. Physician's approval to participate was obtained for all of the older volunteers. Sample size estimates were calculated a priori to detect significant differences in fatigue (% initial torque) with 80% power, an alpha level of 0.05, and an equal number of participants in young and older groups (GPower v3.1; [45]). For this calculation, values from a study that used a similar fatigue protocol were used to determine the effect size [5]. These calculations revealed 8 participants were required per group to detect significant age-related differences in muscle fatigue. Thus, 8 young (21–36 years) and 8 older (66–76 years)

women were studied. To avoid any potential age-by-sex interactions, we restricted this study to women. Participants reported being relatively sedentary ($\leq$ two 30-min bouts of structured exercise per week). They also reported being healthy, as evaluated by a health-history questionnaire, and were not taking any medications known to affect physical function or muscle fatigue (e.g., beta-blockers, calcium channel blockers, etc.). All participants answered 'no' to all questions on the Physical Activity Readiness Questionnaire [46]. To minimize the effect of fluctuating hormone levels on muscle function over the course of the menstrual cycle [47], all young women were studied during days 1–5 of the menses. No participants had used any hormone replacement therapies in the 6 months prior to enrolling in the study.

## Procedures

Participants reported to the Muscle Physiology Laboratory for 2 Visits. At Visit 1, mobility function was measured and familiarization with the fatigue protocol was provided. During Visit 2, muscle architecture of the vastus lateralis was measured, followed by the knee extensor fatigue protocol. Visits were separated by at least 2 days.

## Mobility function and activity

Measures of mobility and physical activity were obtained in order to characterize the study groups. Mobility was measured using the timed up-and-go (TUG; [48]) and advanced short physical performance battery (SPPB-A; [49]). The fastest time for the TUG was used for analysis. At the end of Visit 1, participants were asked to wear a uniaxial accelerometer (GT3X, Actigraph, Pensacola, FL) at the hip for 7 days in order to characterize habitual physical activity levels. Participants wore the accelerometer for a minimum of 10 hours per day, and at least 4 days (3 weekdays and 1 weekend day). These criteria have been shown to provide good reliability (>80%) for quantifying overall and moderate-to-vigorous physical activity [50, 51]. Average daily activity counts and minutes spent in moderate-to-vigorous physical activity were calculated using ActiLife v6.13 software (ActiGraph, Pensacola, FL) with established cut-point thresholds [52].

## Muscle architecture

Thigh, shank, and total leg length were measured using a tape measure. The midpoint between the greater trochanter and lateral femoral condyle of the femur was marked with indelible ink and thigh circumference was measured at this position. Subsequently, the participant was seated on a Biodex System 3 dynamometer (Biodex Medical Systems, Shirley, NY) with hip and knee angles of 90˚ and 100˚, respectively. After 10 min of seated rest to account for fluid shifts in the thigh [53], ultrasound images of the participant's vastus lateralis were collected via a linear array probe (Philips model L12-5, 12-5MHz), using a Philips HD11XE system (Philips Healthcare, Bothell, WA). The probe was placed over the mark on the participant's thigh, and once the probe placement was optimized for visualization of the deep and superficial aponeuroses and muscle fascicles, 5 images were acquired. To check whether peripheral edema occurred during the imaging session, the widest circumference of the calf was measured before and after images were obtained.

Ultrasound images were saved and transferred to a desktop computer for offline analysis using ImageJ software (v1.50i, National Institutes of Health, USA). The pennation angle was identified as the angle between the fascicle and the deep aponeurosis. Muscle thickness was measured as the distance between the internal borders of the superficial and deep aponeuroses. Because visualization of entire fascicles was not possible, fascicle length was estimated as

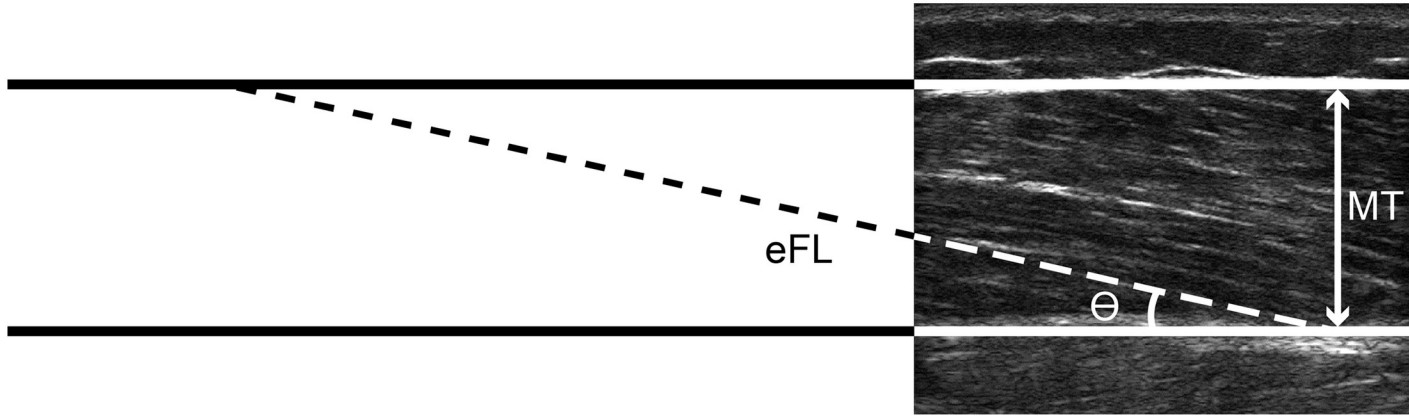

**Fig 1. Ultrasound image from a representative young woman showing estimated fascicle length (eFL), muscle thickness (MT), and pennation angle (Θ).**

follows:

$$\text{Estimated fascicle length} = \frac{\text{Muscle thickness}}{\text{Sin}(\theta)} \qquad (1)$$

where Θ represents the measured pennation angle. This approach, illustrated in Fig 1, is commonly used in studies of muscle architecture [54]. To account for anatomical differences between participants, we report estimated fascicle length in addition to normalizing this measure to thigh length. Analysis of some images was not possible because the deep and superficial aponeuroses were not parallel with each other. As a result, muscle architecture was analyzed for 3 out of the 5 ultrasound images obtained from each participant.

## Muscle torque

Prior to the measurement of isometric and isokinetic torque, participants completed 5 min of unloaded cycling at ~60-rpm to warm-up. The dynamometer was used to measure torque during maximal voluntary isometric (MVIC) and dynamic (MVDC) contractions of the dominant thigh (preferred kicking leg). Participants were seated upright with their hips at 90˚ and a knee angle of 100˚ extension (180˚ = full extension).

Peak knee extensor torque was measured with the dynamometer in isometric and isokinetic modes to determine MVIC and MVDC at $240°\cdot s^{-1}$, respectively. To determine the target torque for electrically-stimulated contractions (see *Myoelectric stimulation* below), participants completed 2 brief MVICs at a knee angle of 100˚ with 1 min rest between contractions. If those 2 MVICs differed by more than 10%, additional MVICs were completed until the 2 strongest contractions were within 10% of each other. To ensure participants were still familiar with the dynamic contractions performed at visit 1, participants completed two sets of 2 MVDCs at a concentric velocity of $240°\cdot s^{-1}$, with 1 min rest between sets to prevent muscle fatigue. Participant's limbs were allowed to return to the start position at a velocity of $500°\cdot s^{-1}$ (i.e., against no load), and all MVDCs were completed over a 70˚ range of motion. For all contractions, the participant sat with her arms folded across her chest, Velcro straps across her torso and hips to prevent unwanted movement, and were instructed to "kick your leg out as hard and fast as possible." Visual torque-feedback, scaled to each participant's maximum torque, was provided during each contraction.

Analog signals corresponding to torque, velocity, and position were acquired from the dynamometer at a sampling rate of 2,500 Hz. Data were analyzed as reported previously [1, 5, 41, 55]. Briefly, peak torque was taken during the target isovelocity period.

## Myoelectric stimulation

Percutaneous electrical stimulation was used to measure muscle contractile properties during isometric contractions at baseline, immediately after the fatigue protocol, and at 5, 10, 20, and 30 min of recovery. Two 7.6 × 12.7 cm self-adhesive stimulating electrodes were placed transversely across the thigh. One electrode was positioned 3–5 cm distal to the inguinal crease and the other was placed 2–3 cm proximal to the superior border of the patella. High- (80-Hz) and low-frequency (10-Hz) stimulations were delivered (200-μs pulse duration, 500 ms) with a constant-current stimulator (DS7AH; Digitimer, Hertfordshire, UK). The current that elicited an 80-Hz contraction resulting in 50% of the participant's MVIC was used for all subsequent stimulations, as described previously [5, 55]. Immediately prior to the fatigue protocol, participants completed a series of voluntary and electrically-stimulated contractions consisting of 2 MVDCs ($240°·s^{-1}$), 1 MVIC, one 80-Hz and one 10-Hz contraction. A representative torque trace from the baseline series of contractions for 1 young woman is shown in Fig 2. This series of contractions was repeated at the end of the fatigue protocol and throughout recovery.

Muscle contractile properties were quantified as peak torque during the 10 and 80-Hz contractions, as well as the maximal rate of torque development (RTD), and $T_{1/2}$ from the 80-Hz tetanus. The RTD was expressed as the percent of peak torque per millisecond ($%pk·ms^{-1}$), to account for the effects of peak torque achieved on this measure. The $T_{1/2}$ was measured as the time for torque to decline to 50% of that achieved at the time of the last stimulus during the 80-Hz contraction. Changes in the ratio of peak torques elicited during the 10-Hz and 80-Hz stimulations was used to determine the presence of LFF [31]. Additionally, the ratio of MVIC:80-Hz torque was calculated at each time point. A decrease in this ratio reflects a greater fall in voluntary than stimulated torque production, suggesting the presence of central fatigue [56].

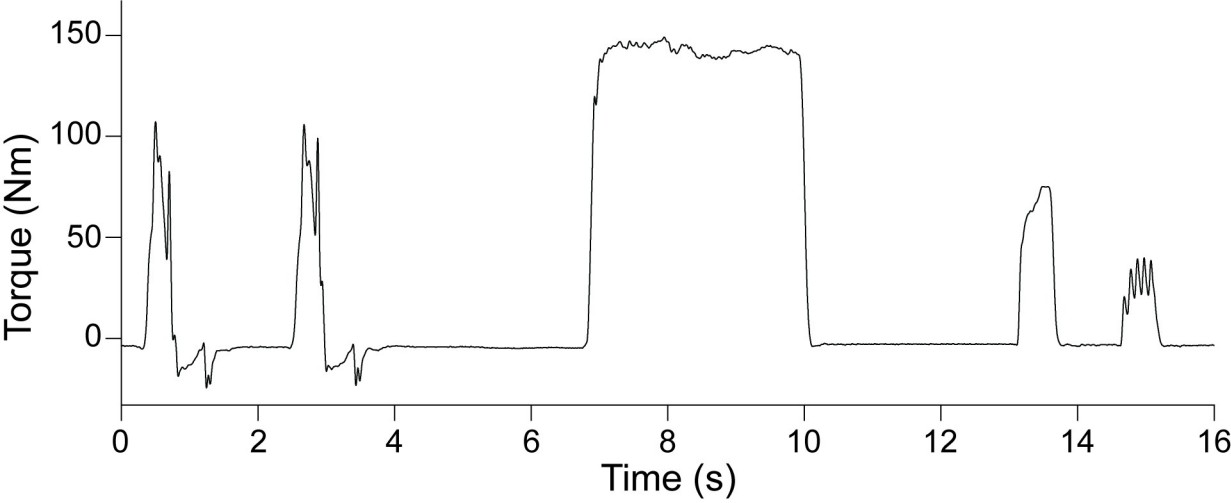

**Fig 2. Representative baseline series of contractions.** Data are from one young woman.

## Fatigue protocol

Following the baseline test series of contractions, the participant completed a unilateral knee extension fatigue protocol consisting of 1 MVDC at $240°·s^{-1}$ every 2 s for 4 min. Each contraction was cued by an auditory signal and strong verbal encouragement was provided by the investigators throughout the protocol. Visual torque feedback, scaled to each participant's MVIC, was provided throughout the fatigue protocol and recovery. Fatigue was calculated as [5]:

$$\frac{\text{(average peak torque of the final 5 contractions)}}{\text{(average of peak torque at baseline and during the fatigue trial)}} \times 100 \qquad (2)$$

To evaluate potential sites of fatigue and the recovery of MVDC torque following fatigue, the test series was repeated immediately after the final contraction of the fatigue trial, and at 5, 10, 20, and 30 min of recovery.

## Statistical analyses

All data were checked for normality and homogeneity of variance using the Kolmogorov-Smirnov test and Levene's statistic, respectively. If the assumptions of normality or homogeneity of variance were violated, non-parametric statistical comparisons were performed. Differences in group descriptive characteristics, muscle architecture, and fatigue were analyzed using independent t-tests or Mann-Whitney U tests. Changes in peak MVDC torque, MVIC:80-Hz torque, 10:80-Hz torque, and contractile properties (RTD and $T_{1/2}$) between baseline and fatigue, and throughout recovery, were analyzed using separate 2-way repeated measures ANOVAs (group × time). Linear regression analyses were performed to evaluate the relationships between muscle architecture and baseline MVIC and MVDC torque, as well as fatigue and measures of physical activity and function, muscle architecture, and contractile properties. For regression analyses in which estimated fascicle length was used, only the non-normalized values were included. All statistical analyses were performed using the SPSS statistical software package (v25.0, IBM, Chicago, IL) with an alpha level of 0.05. Data are reported as mean±SE, with exact p-values, 95% confidence intervals for differences between group means, and effect sizes also provided. Effect sizes were calculated as Cohen's d, where a *d* of 0.2, 0.5, and 0.8 are indicative of small, medium, and large effect sizes [57].

## Results

### Baseline characteristics

Group characteristics are reported in Table 1. The young and older groups had similar height, body mass, and BMI. The young group had greater habitual physical activity (daily counts and MVPA) compared with the older group. The measures of mobility function indicated a difference only for the SPPB-A, which was lower in the older group.

At baseline, older women were weaker than young women (Table 2). There was no difference between groups in the maximum RTD, but $T_{1/2}$ was slower in older compared with young (Table 2).

The within-subject coefficient of variation for the 3 images analyzed for pennation angle, muscle thickness, and estimated fascicle length were 5.7±1.2 and 5.0±1.1, 3.9±1.3 and 3.8±0.9, and 5.7±1.5 and 5.2±1.3% for young and older women, respectively, indicating that these were robust measurements in both groups. Calf circumference did not change from pre- to post-imaging in young (38.8±1.6 vs. 38.8±1.5, p = 0.991) or older (36.4±1.4 vs. 36.5±1.4, p = 0.950) women, suggesting that fluid shifts did not impact the architecture measures. There were no

**Table 1. Group characteristics.**

|  | Young (n = 8) | Older (n = 8) | 95% CI | P | Effect size |
|---|---|---|---|---|---|
| Age (yr) | 23.4 ± 1.8 | 69.6 ± 1.1 | -50.8, -41.6 |  |  |
| Height (cm) | 164.5 ± 1.9 | 164.3 ± 2.6 | -6.6, 7.1 | 0.942 | 0.03 |
| Body mass (kg) | 65.3 ± 5.9 | 66.4 ± 3.8 | -16.2, 14.0 | 0.877 | 0.08 |
| BMI (kg·m$^{-2}$) | 24.1 ± 2.2 | 24.6 ± 1.3 | -6.0, 5.0 | 0.851 | 0.10 |
| PA (counts·d$^{-1000}$) | 286.0 ± 29.2 | 162.6 ± 10.4 | 52.8, 193.8 | 0.005 | 1.99 |
| MVPA (min·d$^{-1}$) | 47.3 ± 6.5 | 11.2 ± 2.3 | 20.5, 51.8 | 0.001 | 2.62 |
| Gait speed (m·s$^{-2}$) | 1.20 ± 0.04 | 1.17 ± 0.07 | -0.1, 0.2 | 0.740 | 0.18 |
| Chair rise time (s) | 14.1 ± 1.1 | 19.5 ± 1.6 | -9.5, -1.3 | 0.140 | 1.41 |
| TUG (s) | 8.3 ± 0.2 | 9.0 ± 0.6 | -2.1, 0.7 | 0.382 | 0.59 |
| SPPB-A | 2.9 ± 0.1 | 2.5 ± 0.1 | 0.02, 0.6 | 0.037 | 1.57 |

Values are mean±SE. 95% CI, 95% confidence intervals for the difference between group means; BMI, body mass index; PA, physical activity; MVPA, moderate- to vigorous-intensity physical activity; TUG, timed up-and-go; SPPB-A, advanced short physical performance battery.

differences between groups in pennation angle, muscle thickness, or fascicle length (Table 2), nor was peak MVDC torque at baseline associated with fascicle length in young ($r^2$ = 0.098, p = 0.449) or older women ($r^2$ = 0.148, p = 0.393). Neither baseline peak MVDC torque nor maximal isometric torque were associated with pennation angle or muscle thickness in young women ($r^2 \leq 0.083$, $p \geq 0.49$). Likewise, neither peak MVDC torque nor maximal isometric torque were associated with pennation angle or muscle thickness in older women ($r^2 \leq 0.143$, $p \geq 0.403$). Lastly, muscle thickness was positively associated with maximal isometric torque in older women ($r^2$ = 0.613, p = 0.022).

## Muscle fatigue

Changes in peak torque during the 4-min fatigue protocol are shown in Fig 3. With the exception of 1 older woman, who could not attain target velocity after the first 25 contractions, all participants were able to complete the 4-min fatigue protocol. Therefore, the fatigue data reported here represent the average for 7 older women. Fatigue was greater in older compared with young women (33.6±4.7 vs. 55.2±4.2% initial torque, respectively; p = 0.004, 95% CI: 8.0–35.1, d = 1.85). Analysis of changes in the MVIC:80-Hz torque ratio (from 2.02±0.06 and

**Table 2. Effects of aging on baseline properties of the vastus lateralis.**

|  | Young (n = 8) | Older (n = 8) | 95% CI | P | Effect size |
|---|---|---|---|---|---|
| *Muscle architecture* |  |  |  |  |  |
| Pennation angle (˚) | 13.5 ± 0.4 | 13.0 ± 0.6 | -1.1, 2.1 | 0.524 | 0.33 |
| MT (mm) | 26.1 ± 1.5 | 23.0 ± 1.8 | -2.0, 8.1 | 0.209 | 0.66 |
| eFL (mm) | 113.7 ± 8.8 | 102.7 ± 8.0 | -14.5, 36.5 | 0.371 | 0.46 |
| eFL/thigh length | 3.1 ± 0.3 | 2.7 ± 0.2 | -0.4, 1.1 | 0.330 | 0.50 |
| *Contractile properties* |  |  |  |  |  |
| MVDC (Nm) | 84.0 ± 7.1 | 49.2 ± 6.0 | 14.9, 54.7 | 0.002 | 1.87 |
| MVIC (Nm) | 134.7 ± 12.0 | 93.8 ± 6.5 | 11.7, 70.3 | 0.01 | 1.50 |
| RTD (%pk·ms$^{-1}$) | 1.33 ± 0.12 | 1.08 ± 0.10 | -0.1, 0.6 | 0.119 | 0.83 |
| $T_{1/2}$ (ms) | 118.8 ± 4.4 | 151.9 ± 6.0 | -49.0, -17.2 | 0.001 | 2.20 |

Values are mean±SE. 95% CI, 95% confidence intervals for differences in group means; MT, muscle thickness; eFL, estimated fascicle length; MVDC, maximal voluntary dynamic contraction; MVIC, maximal voluntary isometric contraction; RTD, maximum rate of torque development; $T_{1/2}$, torque half-relaxation time.

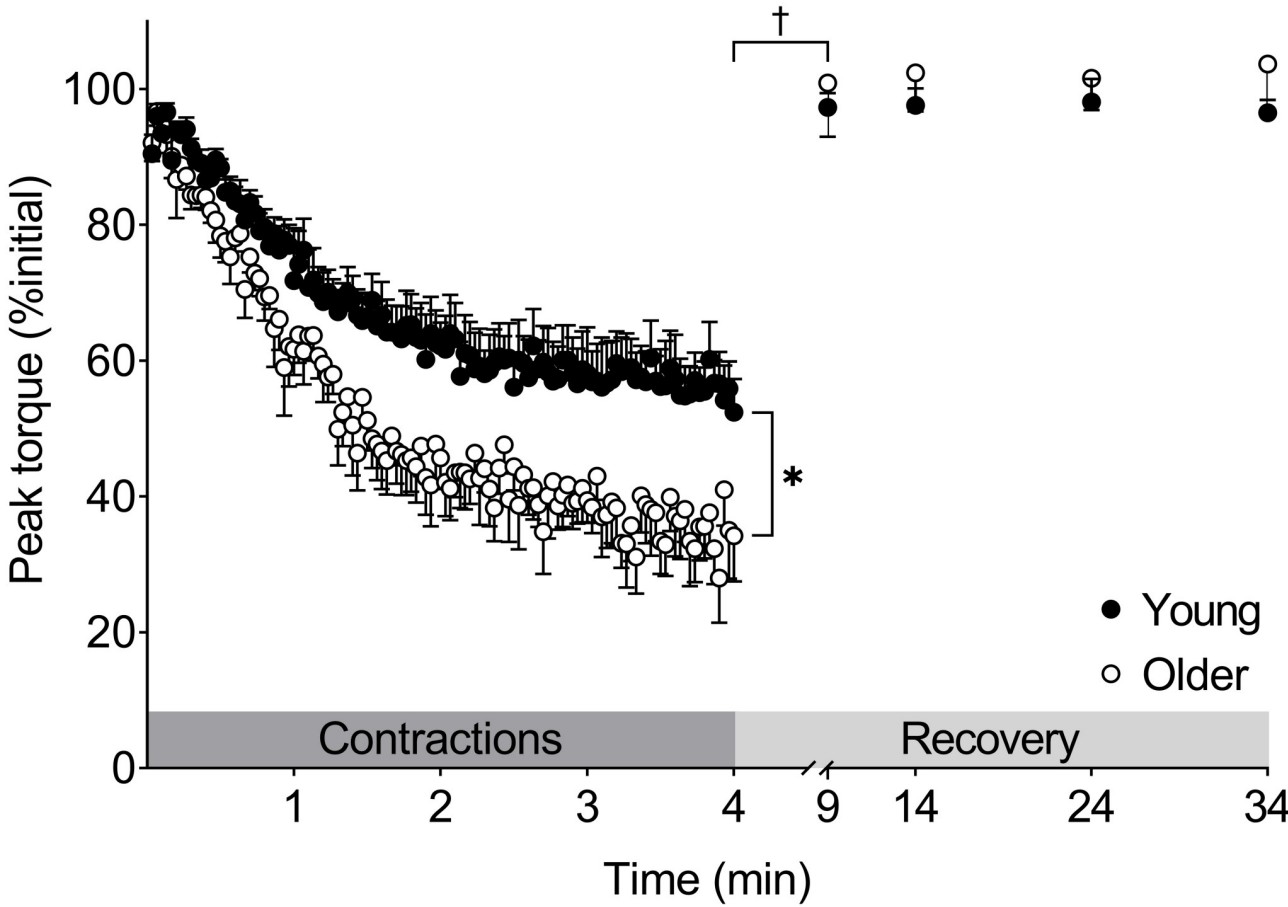

**Fig 3. Age-related differences in muscle fatigue and recovery.** Older women experienced greater fatigue (*p = 0.005). There was a group×time interaction (†p = 0.03) for recovery of peak torque over the first 5 min, but both groups fully recovered within 5 min following contractions. Data are mean±SE. The dark and light grey shaded boxes represent the contraction protocol and recovery period, respectively.

1.87±0.11 at baseline in young and older groups, respectively, to 1.94±0.10 and 1.77±0.09 in young and older at fatigue) revealed no age×time interaction (p = 0.918) or main effects of group (p = 0.097) or time (p = 0.317), indicating no failure of voluntary activation in either group in response to fatiguing contractions.

There was no group×time interaction (p = 0.982) or main effect of group (p = 0.399) for changes in the 10:80-Hz ratio from pre- to post-fatigue, but we did observe a main effect of time (p<0.001) for this measure (Fig 4A), indicating the presence of LFF in both groups. There was no group×time interaction (p = 0.536), but we did observe main effects of group (p = 0.004) and time (p = 0.001) on $T_{1/2}$ such that relaxation slowed similarly with fatigue in both groups (p = 0.001, Fig 4B). The RTD did not change with fatigue in young (p = 0.783) or older groups (p = 0.99, S1 Table).

### Recovery from fatigue

The recovery of peak MVDC torque showed a group×time interaction (p = 0.03) such that older women recovered more in the first 5 min following the protocol than young women (p = 0.001; Fig 3). Peak MVDC torque did not differ from baseline by 5 min of recovery in either group. Maximal isometric torque recovered following the fatigue protocol (main effect

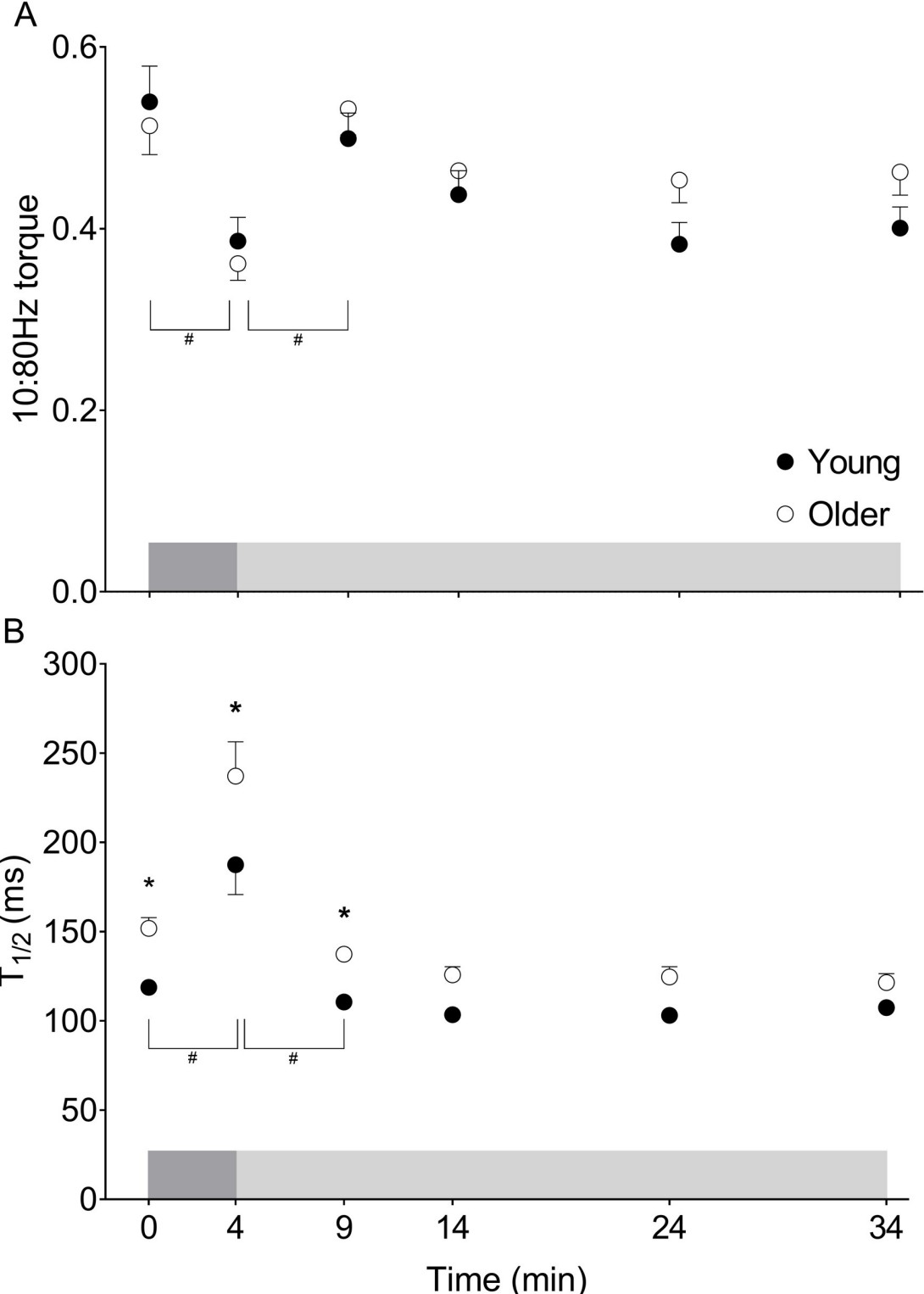

**Fig 4. Fatigue and recovery of contractile properties in young and older women.** A) 10:80-Hz torque at baseline, fatigue, and throughout recovery. The 10:80-Hz torque ratio decreased with fatigue (main effect of time: #p = 0.001) and increased over the first 5 minutes of recovery in both groups (main effect of time: #p≤0.01). B) $T_{1/2}$ at baseline, fatigue, and throughout recovery. Relaxation was slower in older compared with young women at baseline, fatigue, and following 5 minutes of recovery (main effect of group: *p≤0.04). In both groups, $T_{1/2}$ slowed with fatigue and recovered within 5 minutes of recovery in both groups (main effect of time: #p = 0.001), with no group×time interactions. Data are mean±SE. The dark and light grey shaded boxes represent the contraction protocol and recovery period, respectively.

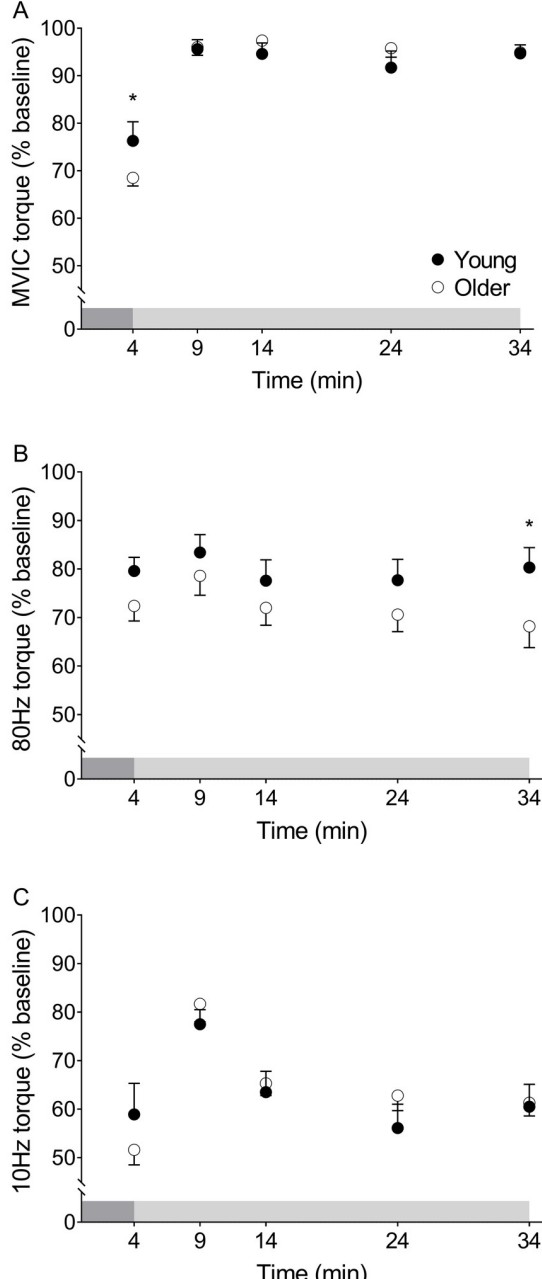

**Fig 5. Recovery of maximal voluntary isometric and stimulated torque measures in young and older women.** A) Maximal voluntary isometric torque throughout recovery. Torque decreased less in young than older adults in response to fatigue, but both groups recovered within 5 min. B) 80-Hz torque throughout recovery. The torque elicited by an 80-Hz stimulation was only different after 30 min recovery, such that it was greater in young than older adults. C) 10-Hz torque throughout recovery. No group differences were observed at any time point.

of time, $p = 0.001$), with no effect of group ($p = 0.95$) and no group×time interaction ($p = 0.103$, Fig 5). Full recovery of MVIC was observed 5 min post-fatigue in young and older groups ($p \leq 0.001$).

The 10:80-Hz torque ratio revealed no group×time interaction ($p = 0.326$), but we did observe main effects of time ($p < 0.001$) and group ($p = 0.037$, Fig 4A) during recovery. The

10:80-Hz torque ratio increased to pre-fatigue levels within 5 min of recovery in both groups ($p \leq 0.01$). There was no group×time interaction ($p = 0.359$), but we did observe main effects of time ($p < 0.001$) and group ($p < 0.001$) for changes in $T_{1/2}$ during recovery such that torque relaxation recovered similarly in young and older women (Fig 4B). There was no group×time interaction ($p = 0.973$) and no main effects of group ($p = 0.085$) or time ($p = 0.867$) for the RTD during recovery (S1 Table).

### Factors associated with fatigue

Regression analyses revealed that fatigue was not associated with any parameter of muscle architecture in the young or older women ($r^2 \leq 0.14$, $p \geq 0.462$). The ΔMVIC:80-Hz torque from baseline to post-fatigue was not associated with fatigue in young ($r^2 = 0.17$, $p = 0.305$) or older women ($r^2 = 0.001$, $p = 0.995$), confirming that voluntary activation failure did not contribute to fatigue in this study. The Δ10:80-Hz torque from baseline to post-fatigue was directly associated with fatigue in young ($r^2 = 0.713$, $p = 0.008$) but not older women ($r^2 = 0.012$, $p = 0.815$). In contrast, baseline $T_{1/2}$ was inversely associated with fatigue in older ($r^2 = 0.584$, $p = 0.045$) but not young women ($r^2 = 0.147$, $p = 0.348$), as illustrated in Fig 6. Neither physical activity counts nor MVPA were associated with fatigue in young ($r^2 \leq 0.034$, $p \geq 0.661$) or older ($r^2 \leq 0.053$, $p \geq 0.618$) women. Fatigue was associated with SPPB-A score in older ($r^2 = 0.565$, $p = 0.051$), but not younger ($r^2 = 0.032$, $p = 0.673$) women.

## Discussion

This study was designed to evaluate whether age-related differences in muscle architecture, voluntary activation, or contractile properties might be potential mechanisms for the greater muscle fatigue observed in older adults in response to high-velocity muscle contractions. Peak MVDC torque fell significantly more in healthy older women than in healthy young women, demonstrating the expected age-related difference in knee extensor muscle fatigue at high contraction velocities. The results indicate that baseline differences in contractile properties, and not acute changes in these properties in response to fatigue, may contribute to high-velocity fatigue in aging: baseline $T_{1/2}$ was slower in older than young, and predictive of fatigue only in the older group. There were no age-related differences in muscle architecture across our study groups, and we provide new information that architecture did not contribute to the greater fatigue observed in the older group. Comparisons of the fall in voluntary vs. stimulated tetanic contractions during fatigue (i.e., MVIC:80Hz torque ratio) indicated no decrement in voluntary activation in either group in response to this high-velocity contraction protocol, ruling out activation failure as a source of the difference in muscle fatigue between age groups. Comparable LFF was observed in both groups, indicating significant peripheral fatigue in response to this protocol that was independent of age. Overall, these results support previous reports of greater fatigue in older muscle during high-velocity work [5, 24, 29], and suggest that this difference may be due in part to chronic changes in muscle properties, but not architecture, in the aged.

### Baseline

Notably, there were no age-related differences in baseline muscle architecture of the vastus lateralis in these groups of healthy women (Table 2). While some studies have reported remodelling of muscle architecture in old age [4, 20–22] such that muscle thickness, pennation angle, and fascicle length are reduced with age, most of those studies were conducted in men. In contrast to those results, our data are in agreement with those of Kubo et al. [58] who likewise observed no age-related differences in vastus lateralis fascicle length, normalized to thigh

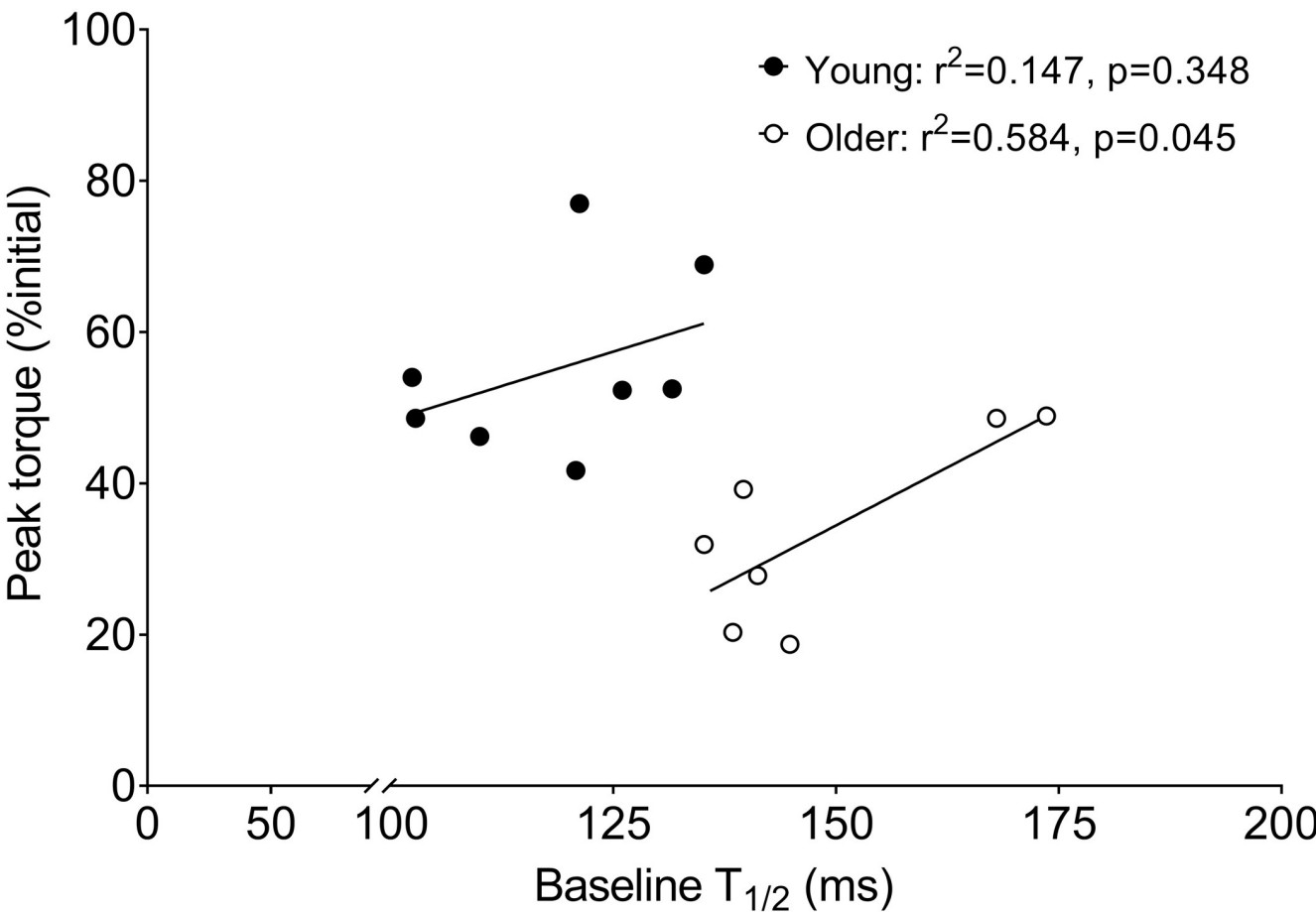

**Fig 6. Associations between half relaxation time and fatigue.** Fatigue (peak torque, % initial) was plotted against half relaxation time ($T_{1/2}$) at baseline. Linear regression analysis revealed an association between these variables for older ($r^2$ = 0.584, p = 0.045), but not young women ($r^2$ = 0.147, p = 0.348), suggesting the age-related slowing of $T_{1/2}$ may prevent excessive muscle fatigue in older women.

length, in women. However, our results differ from that study insofar as we observed no age-related differences in pennation angle and muscle thickness. The cause of this discrepancy is not clear at present, but does not appear to be due to differences in the age of the participants, nor the repeatability of the measurement given the low coefficient of variation found in our study. Questions as to how factors such as habitual physical activity and sex may influence architectural changes in old age, as well as the potential functional impacts of these changes, remain to be answered.

At baseline, peak MVDC and MVIC torque were ~40 and ~30% lower, respectively, in older compared with young women (Table 2), which is consistent with previous reports [5, 55, 59]. The maximal RTD during electrically-evoked contractions was not different between young and older groups, similar to some [5, 55] but not all reports in the literature [29]. The $T_{1/2}$ was slower at baseline in older compared with young muscle, a result that is in agreement with most studies [5, 24, 25, 55], and potentially due to slowed $Ca^{2+}$ dissociation from troponin, slower crossbridge kinetics [40], a greater relative proportion of type I fibers [9], or a decrease in muscle-tendon unit stiffness with age [60]. The slower relaxation in older muscle does not appear to be caused by slowed $Ca^{2+}$ re-uptake by the sarcoplasmic reticulum with age [61].

## Fatigue

Consistent with previous reports [5, 24, 25, 29], we observed greater knee extensor muscle fatigue in older compared with young women during contractions at $240°·s^{-1}$. The question of whether older muscle fatigues more than young depends upon the contraction velocity used to fatigue the muscle, with greater fatigue reported at higher speeds, i.e., those to the right on the torque-velocity curve [5]. The fatigue resistance of older muscle under isometric conditions is well reported [28], and the cause of this difference appears to be due to a lower reliance on glycolytic ATP production in older compared with young muscle [62, 63]. Fatigue induced by slow- and moderate-velocity dynamic contractions is similar between young and older adults [5, 24, 55], while fatigue is generally greater in older than young muscle in response to high-velocity contractions [5, 24–26, 29, 44], as found here. The goal of this study was to evaluate several candidate mechanisms for this difference in fatigue at high contraction velocities.

Although the young women in our study were more physically active than the older women (Table 1), the magnitude of fatigue in each group was nearly identical to that reported by Callahan and Kent-Braun [5], who matched age groups for physical activity and used a similar fatigue protocol to the one used here. Moreover, Englund et al. [64] reported that 12 weeks of progressive resistance training in mobility-limited older men and women improved total torque output (i.e., strength) during a moderate-velocity fatigue protocol of the knee extensors, but had no effect on fatigue, expressed as the relative fall in peak torque. The effects of an intervention designed specifically to increase muscular endurance (i.e., fatigue resistance) remain to be determined, but the results to date suggest that the greater fatigue in older compared with young adults during high-velocity contractions may be independent of physical activity status. In further support of this notion, we observed no relationship between physical activity counts or MVPA and fatigue in either young or older women.

To evaluate the potential role of changes in the completeness of voluntary activation on the age-related differences in muscle fatigue observed here, we compared the changes in the MVIC:80-Hz torque from baseline to fatigue [56]. There were no changes in this measure from baseline to fatigue in either group, indicating no decrease in voluntary activation with fatigue due to high-velocity contractions, a result similar to that reported previously in men [24]. Recently, transcranial magnetic stimulation and femoral nerve stimulation were used to examine potential neural mechanisms for greater fatigue with age following high-velocity contractions of the knee extensors in men and women [29]. In that study, the authors observed a reduction in voluntary activation with fatigue in older, but not young women, but these changes in voluntary activation were small and not associated with fatigue. Rather, the authors found a positive association between fatigue and the decrease in stimulated twitch torque, consistent with an age-related difference in peripheral fatigue due to impairment within the muscle, potentially including failure of ECC or crossbridge function. Indeed, subsequent work by this group has shown a significant association between metabolic by-products (Pi, $H^+$, $H_2PO_4^-$) and fatigue during a similar protocol, supporting the notion that age-related differences in muscle fatigue occur to differences in the intracellular milieu (62).

Based upon measures of the m-wave (EMG response to a single electrical stimulus), previous studies of knee extensor muscle fatigue indicate that impaired excitability of the neuromuscular junction or along the sarcolemma are not responsible for the age-related differences observed in fatigue during voluntary dynamic contractions such as used here, including those at high velocities [24, 29]. Thus, these studies also suggest that fatigue under these conditions originates within the muscle. To determine whether greater fatigue in older compared with young muscle was due to failure of ECC, i.e., from the t-tubules to the cross-bridges, we compared changes in the 10:80-Hz torque ratio from baseline to fatigue [31]. This ratio decreased

similarly with fatigue in both groups, indicating a comparable failure of excitation-contraction coupling in both young and older women. Thus, it appears that excitation-contraction coupling failure does not explain the age-group difference in fatigue in this study.

Torque relaxation ($T_{1/2}$) in response to the 80-Hz tetanic stimulus slowed to a similar extent in young and older women during fatigue, which is consistent with some literature [33, 41]. In contrast, a greater slowing of twitch $T_{1/2}$ in older compared with young muscle following moderate- and high-velocity, but not slow-velocity contractions of the knee extensors has been reported [24], with no sex-based differences observed [29]. The slowing of $T_{1/2}$ with fatigue is likely due to an accumulation of proton in the cytosol during contractions [43], which has been shown to inhibit $Ca^{2+}$-ATPase ~2-fold with a change in pH from 7.1 to 6.6 [65]. Indeed, recent work has shown the greater fatigue in older compared with young adults is due to more accumulation of Pi and $H^+$ in the knee extensor muscles of older adults [44].

Notably, baseline $T_{1/2}$ was predictive of fatigue in older, but not young women, such that ~60% of the variation in fatigue during high-velocity contractions in the older group was associated with the rate of torque relaxation in the unfatigued muscle. Whether this relationship may reflect an underlying alteration to contractile function that is protective against the greater fatigue in older muscles during high-velocity contractions, is not known at present. The mechanisms for slowed $T_{1/2}$ in old age are not entirely clear, but do not appear to include impaired calcium resequestration by the sarcoplasmic reticulum [61]. Theoretically, the age-related slowing of $T_{1/2}$ could be caused by: 1) a relatively greater volume of type 1 muscle fibers, 2) slowed detachment of myosin from actin, and 3) a less stiff muscle-tendon unit. These potential mechanisms require further evaluation in the context of muscle fatigue in aging.

### Recovery from fatigue

Although peak MVDC torque and maximal isometric torque were lower in older compared with young women at fatigue, both groups returned to pre-fatigue values by the fifth minute of recovery (Fig 3). In contrast to our observation of maximal isometric torque recovering within 5 min, Allman and Rice found incomplete recovery of maximal isometric force of the elbow flexor muscles in young and older men following intermittent, submaximal isometric contractions, which likewise was accompanied by low-frequency fatigue in both groups [32]. More recently, Kent-Braun et al. [66] reported no age-related differences in the recovery of torque following 4-min of contractions of the knee extensors at 120˚·s⁻¹. Those authors also observed maximal torque was almost completely recovered within 5 min, consistent with our observations here. Overall, the rapid and complete recovery of MVDC and isometric torque following a high-velocity protocol that induced substantial muscular fatigue suggests a robust resilience of the neuromuscular system in this group of healthy but sedentary older women.

### Conclusions

This study shows that the greater muscle fatigue observed in older compared with young muscle during high-velocity contractions was not related to differences in muscle architecture, deficits in voluntary activation, failure of excitation-contraction coupling, or an exaggerated, fatigue-induced slowing of contractile properties. We have shown that slower torque relaxation at baseline is inversely associated with greater muscle fatigue in older women, which might be explained by selective atrophy of fast-twitch muscle fibers, slowed crossbridge kinetics, or changes in muscle-tendon unit stiffness with age. Overall, our results suggest that the cause of age-related differences in muscle fatigue reside within the myofibers in healthy older adults; recent work suggests that intracellular metabolism may be important in this process [44].

## Supporting information

**S1 Table. Age-related differences in the peak rate of torque development.** Values are mean ±SE. 95% CI, 95% confidence intervals for the difference between group means; 0R, zero recovery (immediately following the fatiguing contraction protocol); 5R, 5-min recovery; 10R, 10min-recovery; 20R, 20-min recovery; 30R, 30-min recovery.
(DOCX)

## Acknowledgments

The authors thank the participants for their contributions to this study. We are also grateful to Dr. Graham E. Caldwell for his insight and discussion relating to the acquisition and analysis of the ultrasound images. Lastly, we thank Sydney L. Connor and Jamie M. Truax for their help with the data analysis.

## Author Contributions

**Conceptualization:** Liam F. Fitzgerald, Jane A. Kent.

**Data curation:** Liam F. Fitzgerald, Jane A. Kent.

**Formal analysis:** Liam F. Fitzgerald, Margaret M. Ryan, Miles F. Bartlett, Jane A. Kent.

**Investigation:** Liam F. Fitzgerald, Margaret M. Ryan, Miles F. Bartlett, Jules D. Miehm.

**Methodology:** Liam F. Fitzgerald, Jane A. Kent.

**Project administration:** Liam F. Fitzgerald.

**Supervision:** Jane A. Kent.

**Visualization:** Liam F. Fitzgerald, Jane A. Kent.

**Writing – original draft:** Liam F. Fitzgerald.

**Writing – review & editing:** Liam F. Fitzgerald, Margaret M. Ryan, Miles F. Bartlett, Jules D. Miehm, Jane A. Kent.

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
