## [Decision Letter · Decision Letter 0]

25 Aug 2020

PONE-D-20-14730

Effects of old age on knee extensor fatigue and recovery from high-velocity muscle contractions

PLOS ONE

Dear Dr. Fitzgerald,

Thank you for submitting your manuscript to PLOS ONE. After careful consideration, we feel that it has merit but does not fully meet PLOS ONE’s publication criteria as it currently stands. Therefore, we invite you to submit a revised version of the manuscript that addresses the points raised during the review process.

We look forward to receiving your revised manuscript.

Kind regards,

Dragan Mirkov, Ph.D.

Academic Editor

PLOS ONE

Journal Requirements:

Reviewers' comments:

Reviewer's Responses to Questions

**Comments to the Author**

1. Is the manuscript technically sound, and do the data support the conclusions?

Reviewer #1: Yes

Reviewer #2: Partly

2. Has the statistical analysis been performed appropriately and rigorously? 

Reviewer #1: Yes

Reviewer #2: Yes

3. Have the authors made all data underlying the findings in their manuscript fully available?

Reviewer #1: Yes

Reviewer #2: Yes

4. Is the manuscript presented in an intelligible fashion and written in standard English?

Reviewer #1: Yes

Reviewer #2: Yes

5. Review Comments to the Author

Reviewer #1: Overall, this is a clear, concise, and well-written manuscript. The introduction is relevant and theory based. Sufficient information about the previous study findings is presented for readers to follow the present study rationale and procedures. The methods are generally appropriate. In essence, the methods part is too long. I suggest simplifying it. Also, this applies to results part. However, results are well presented. Discussion is appropriate, mainly regarding the implications of the findings of the study.

Reviewer #2: The aim of this research paper was to examine the potential roles of resting muscle architecture and baseline contractile properties, as well as changes in voluntary activation and low-frequency fatigue in response to high-velocity knee extensor work. While the authors purposefully included only female participants to avoid gender-related interactions, I believe it would be of interest to compare them with males particularly since the previous results are ambiguous. Also, while it is good that you used G power to determine sample size, why not including at least 10 subjects instead of only 8 as suggested,just because of the possibility of someone dropping out? Further, the authors might want to consider reviewing some studies that investigated the association of RFD and aging and the implications of the most rapid isometric contractions. Contractile RFD might be equally if not more sensitive to fatigue than power.

The authors put a great effort both in lab work and writing. The introduction is very extensive and somewhat hard to follow, especially the last paragraph. Methods are very detailed and followed by appropriate statistical analysis. Results and discussion detailed and extensive and similarly as introduction, sometimes hard to follow. The conclusions should provide the bottom-line of your hard work, but I don’t find it. Outline the novelty and the significance.

Specific comments:

Title: It seems that something like “Muscle architecture, voluntary activation, and low-frequency fatigue do not explain the greater fatigue of older compared with young women during high-velocity contractions“ would fit better to what you presented here

Line 54: A number of studies have shown that RFD rather than power has association with such risk

Line 61: And sarcopenia that kicks-in by the age of 40.

Line 184 and 209: Can you please justify positioning with the “hips at 90° and a knee angle of 100° extension (180° = full extension)”? Why not using 120 degrees in knee extension?

L230: What was the reason for sampling at 2500 Hz? Please provide a reference for data acquisition.

L251: Why did you chose peak RTD and not “contractile RTD” as suggested by Aagaard et al 2002, 2007 etc.? RTS derived at various time intervals from the contraction onset has been shown as sensitive to both training and surgical interventions.

L283: Please provide the interpretation for Cohen’s d

Table 3: Consider presenting these data as three line-graphs.

6. PLOS authors have the option to publish the peer review history of their article (what does this mean?). If published, this will include your full peer review and any attached files.

Reviewer #1: **Yes: **Patrik Drid

Reviewer #2: No

---

## [Author Response · Author response to Decision Letter 0]

7 Oct 2020

Reviewer #1: Overall, this is a clear, concise, and well-written manuscript. The introduction is relevant and theory based. Sufficient information about the previous study findings is presented for readers to follow the present study rationale and procedures. The methods are generally appropriate. In essence, the methods part is too long. I suggest simplifying it. Also, this applies to results part. However, results are well presented. Discussion is appropriate, mainly regarding the implications of the findings of the study.

Thank you for your enthusiasm in our manuscript. Based on your feedback, we have revised the methods section to be more concise.

Reviewer #2: The aim of this research paper was to examine the potential roles of resting muscle architecture and baseline contractile properties, as well as changes in voluntary activation and low-frequency fatigue in response to high-velocity knee extensor work. While the authors purposefully included only female participants to avoid gender-related interactions, I believe it would be of interest to compare them with males particularly since the previous results are ambiguous. Also, while it is good that you used G power to determine sample size, why not including at least 10 subjects instead of only 8 as suggested,just because of the possibility of someone dropping out? Further, the authors might want to consider reviewing some studies that investigated the association of RFD and aging and the implications of the most rapid isometric contractions. Contractile RFD might be equally if not more sensitive to fatigue than power.

The authors put a great effort both in lab work and writing. The introduction is very extensive and somewhat hard to follow, especially the last paragraph. Methods are very detailed and followed by appropriate statistical analysis. Results and discussion detailed and extensive and similarly as introduction, sometimes hard to follow. The conclusions should provide the bottom-line of your hard work, but I don’t find it. Outline the novelty and the significance.

Thank you for your careful review of our manuscript. While inclusion of males and females, with a sufficient sample size in each group, might have been interesting, Sundberg et al. observed no sex-related differences in muscle fatigue, recovery of muscular torque or power, or voluntary activation when using a similar fatiguing protocol (Sundberg et al. 2018, JAP). Thus, we are confident that only including women in this study did not affect the generalizability of our results. 

Specific comments:

Title: It seems that something like “Muscle architecture, voluntary activation, and low-frequency fatigue do not explain the greater fatigue of older compared with young women during high-velocity contractions“ would fit better to what you presented here

Thank you for this suggestion. We have changed the title of our manuscript to the one you suggested.

Line 54: A number of studies have shown that RFD rather than power has association with such risk.

Whilst some authors report an association between the rate of torque development (RTD) and fall risk with aging, other authors have reported no age-related differences in the RTD (Callahan et al. 2009, Muscle Nerve; Chung et al. 2007, JAP). One important distinction between RTD and muscular power is that RTD is often measured under isometric conditions, as we did in this study. Perhaps, then, the important component here is the rapid development of muscular force, as we highlight in Lines 58-61.

Line 61: And sarcopenia that kicks-in by the age of 40.

Sarcopenia refers to the progressive loss of muscle mass and the resulting impairment to physical function, with age (Morley et al. 2011, J Am Med Dir Assoc). Baumgartner et al. provided an operational definition of sarcopenia as being 2 standard deviations below the muscle mass of healthy young individuals (Baumgartner et al. 1998, Am J Epidemiol). Although muscle mass declines at ~1% per year after the age of ~40 years, severe loss of muscle (2 standard deviations below young healthy individuals) is only present in 5 to 13% of 60-70 year olds (Fielding et al. 2011, J Am Med Dir Assoc; von Haehling et al. 2010, J Cachexia Sarcopenia Muscle). We are not certain what the reviewer intended with this comment, but we highlight the effects of reduced muscle mass on architectural remodeling in the subsequent paragraph. 

Line 184 and 209: Can you please justify positioning with the “hips at 90° and a knee angle of 100° extension (180° = full extension)”? Why not using 120 degrees in knee extension?

We chose these hip and knee angles for three reasons. Firstly, this is a comfortable sitting position for participants. Secondly, this position has been used previously in studies of age-related differences in muscle fatigue (Callahan et al. 2009, Muscle Nerve; Callahan & Kent-Braun, 2011, JAP). Lastly, we have noted that some older adults cannot reach full extension (180°) whilst sitting due to hamstring tightness. Therefore, a knee angle of 100° allowed participants to perform the contractions over a large range of motion (70°), which would not have been feasible with a more extended starting knee angle. 

L230: What was the reason for sampling at 2500 Hz? Please provide a reference for data acquisition.

We had originally planned to collect EMG data throughout the fatigue protocol (although this did not work out), so we used a high sampling rate on all channels. This approach has been used previously in our lab (Callahan & Kent-Braun, 2011, JAP). Further, others have shown no difference in peak torque or the rate of torque development at sampling frequencies of 500, 1,000, and 2,000Hz (Thompson 2019, J Biomech), so we are confident the sampling frequency did not affect any measures in the current study. 

L251: Why did you chose peak RTD and not “contractile RTD” as suggested by Aagaard et al 2002, 2007 etc.? RTS derived at various time intervals from the contraction onset has been shown as sensitive to both training and surgical interventions.

We reported the peak rate of torque development as a percentage of the peak torque elicited by the 80Hz tetanic contraction, per millisecond (% peak per millisecond; %pk.ms-1) which is consistent with previous work in our lab (Callahan et al. 2009, Muscle Nerve; Chung et al. 2007, JAP; etc). Normalizing to peak torque achieved accounts for the effects of torque production on the rate of torque development (Miller et al, 1981) and thus focuses on the speed of the contraction. Based upon Figure 2 in the paper you refer to (Aagaard et al, 2007, MSSE), our measure of peak RTD is similar to the value they reported at 50ms. Further, Aagaard et al. showed that the contractile RTD was most different between their groups at 50ms. Thus we are confident that we would have observed the same result (i.e., no difference in RTD between young and older at baseline nor with fatigue) if we had reported contractile RTD instead of peak RTD. In response to the reviewer’s comment, we now report the peak RTD throughout recovery in Table S1 (supplementary information). The group data for RTD at baseline are reported in Table 2 of the manuscript.

L283: Please provide the interpretation for Cohen’s d

We have provided qualitative interpretation of different d values on Lines 288-289, based on Cohen’s work. 

Table 3: Consider presenting these data as three line-graphs.

Thank you for this suggestion. We now report these data in Figure 5.

---

## [Decision Letter · Decision Letter 1]

19 Oct 2020

Muscle architecture, voluntary activation, and low-frequency fatigue do not explain the greater fatigue of older compared with young women during high-velocity contractions

PONE-D-20-14730R1

Dear Dr. Fitzgerald,

We’re pleased to inform you that your manuscript has been judged scientifically suitable for publication and will be formally accepted for publication once it meets all outstanding technical requirements.

Kind regards,

Dragan Mirkov, Ph.D.

Academic Editor

PLOS ONE

Additional Editor Comments (optional):

Reviewers' comments:

Reviewer's Responses to Questions

**Comments to the Author**

1. If the authors have adequately addressed your comments raised in a previous round of review and you feel that this manuscript is now acceptable for publication, you may indicate that here to bypass the “Comments to the Author” section, enter your conflict of interest statement in the “Confidential to Editor” section, and submit your "Accept" recommendation.

Reviewer #1: All comments have been addressed

Reviewer #2: All comments have been addressed

2. Is the manuscript technically sound, and do the data support the conclusions?

Reviewer #1: Yes

Reviewer #2: Yes

3. Has the statistical analysis been performed appropriately and rigorously? 

Reviewer #1: Yes

Reviewer #2: Yes

4. Have the authors made all data underlying the findings in their manuscript fully available?

Reviewer #1: Yes

Reviewer #2: Yes

5. Is the manuscript presented in an intelligible fashion and written in standard English?

Reviewer #1: Yes

Reviewer #2: Yes

6. Review Comments to the Author

Reviewer #1: I wish to congratulate the authors for the job done in this manuscript. This was not an easy experiment to conduct. The impact of fatigue following high-velocity concentric contractions presents a serious problem, and research covering this topic deserves attention. In general, the paper is very well written and structured with proper methodology.

Reviewer #2: (No Response)

7. PLOS authors have the option to publish the peer review history of their article (what does this mean?). If published, this will include your full peer review and any attached files.

Reviewer #1: **Yes: **Patrik Drid

Reviewer #2: No

---

## [Editor Report · Acceptance letter]

23 Oct 2020

PONE-D-20-14730R1 

Muscle architecture, voluntary activation, and low-frequency fatigue do not explain the greater fatigue of older compared with young women during high-velocity contractions 

Dear Dr. Fitzgerald:

I'm pleased to inform you that your manuscript has been deemed suitable for publication in PLOS ONE. Congratulations! Your manuscript is now with our production department. 

Kind regards, 

on behalf of

Dr. Dragan Mirkov 

Academic Editor

PLOS ONE